# Long-term deep-supercooling of large-volume water and red cell suspensions via surface sealing with immiscible liquids

Haishui Huang[1], Martin L. Yarmush[1,2] & O. Berk Usta [1]

Supercooling of aqueous solutions is a fundamentally and practically important physical phenomenon with numerous applications in biopreservation and beyond. Under normal conditions, heterogeneous nucleation mechanisms critically prohibit the simultaneous long-term (>1 week), large volume (>1 ml), and low temperatures (< −10 °C) supercooling of aqueous solutions. Here, we report on the use of surface sealing of water by an oil phase to significantly diminish the primary heterogeneous nucleation at the water/air interface. We achieve deep supercooling (down to −20 °C) of large volumes of water (up to 100 ml) for long periods (up to 100 days) simultaneously via this approach. Since oils are mixtures of various hydrocarbons we also report on the use of pure alkanes and primary alcohols of various lengths to achieve the same. Furthermore, we demonstrate the utility of deep supercooling via preliminary studies on extended (100 days) preservation of human red blood cells.

[1] Center for Engineering in Medicine, Massachusetts General Hospital, Harvard Medical School and Shriners Hospitals for Children, Boston, Massachusetts 02114, United States. [2] Department of Biomedical Engineering, Rutgers University, Piscataway, New Jersey 08854, United States. Correspondence and requests for materials should be addressed to M.L.Y. (email: ireis@sbi.org) or to O.B.U. (email: berkusta@gmail.com)

Water is a seemingly simple yet practically complex liquid with extraordinary phase behavior, which enables many of life's intricacies. While water is possibly the most studied liquid, there remain many areas where its behavior is still mysterious[1]. A prime example for this is the freezing and the supercooling of water occurring in our daily lives and scientific research[2,3]. Ice formation and the preceding supercooled state of microdroplets in atmospheric clouds are crucial elements for precipitation and reflection of solar radiation[4,5]. Furthermore, chilling, freezing, freeze avoidance, and supercooling are important strategies to combat cold environment for ectothermic animals[6,7], treat malignant diseases via cryotherapy[8], and preserve food and various biological samples, such as cells, tissues, and organs[9,10].

Recent advances have shown that supercooling can be a promising alternative approach for the preservation of cells, tissues, and especially organs[11]. Nevertheless, an important hurdle for supercooling preservation, as well as other applications of supercooling, is that simultaneous low temperature ($< -10\,°C$), large volume (>1 ml), and long period (>1 week) of supercooling for aqueous solutions cannot be readily achieved[12–14]. High-pressure-based approaches have provided supercooled states of water down to $-92\,°C$ briefly[1], according to the water phase diagram. They are, however, expensive, might further complicate preservation of biological samples, and their long-term fate is unknown. Few experiments have unstably supercooled large volumes, several hundred milliliters, of water to $-12\,°C$[15], albeit also for periods on the order of seconds. Similarly, in Dorsey's classical work on freezing of supercooled water, he was able to achieve a temperature of $-19\,°C$ for a few milliliters of water very briefly during his constant cooling experiments[16]. A method that overcomes these hurdles and enables long-term supercooling of large aqueous samples at low temperatures could find applications in biopreservation, as well as many other areas which have previously been practically prohibited.

Under normal atmospheric conditions, ice melts at $0\,°C$, the ice-water equilibrium temperature ($T_e$). Nevertheless, the observed freezing temperature ($T_f$) for pure water could fall below $T_e$ since successful ice nucleation at $0\,°C$ can take a long time. Water, in the liquid phase, below the equilibrium temperature is said to be "supercooled" where $\Delta T = T_e - T_f$ measures the degree of supercooling. Supercooled water is intrinsically metastable and can spontaneously transform to lower-energy-level ice crystals through the formation of ice nuclei, which can be readily achieved by ice seeding[17], ultrasonicating[18], or presenting ice-nucleating agents[19]. On the contrary, it is very difficult to maintain supercooled water unfrozen, especially for a large volume, under a high degree of supercooling, or for a long period, as each of these increases the possibility of ice nucleation and water freezing (Supplementary Note 1). For instance, $\Delta T$ of a water droplet decreases logarithmically with increasing volume under a constant cooling rate[20]. Similarly, supercooling frequency ($f_s$, $f_s$ = number of unfrozen droplets/number of total droplets) of an ensemble of droplets decreases exponentially with increasing droplet volume, storage time, and nucleation rate ($J$)[21,22], while $J$ itself increases exponentially with $\Delta T$[23]. Consequently, simultaneous long-term (>1 week), large volume (>1 ml), and deep supercooling (DSC) ($\Delta T > 10\,°C$) of water has not yet been achieved.

There are two general ice nucleation mechanisms, homogenous and heterogeneous crystallization. Homogeneous crystallization occurs due to random aggregation of interior water molecules to create a critically large nucleus of ice crystal, which could only be achieved and observed below $-20\,°C$[24]. Heterogeneous crystallization, on the other hand, stems from ice nucleus formation catalyzed by a substrate and/or with the aid of foreign objects at much higher temperatures[25]. Consequently, water freezing is generally initiated by heterogeneous nucleation, and the water/air interface is the primary nucleation site as revealed in theoretical[26,27], experimental[14,28], and numerical[29,30] studies.

Here, we describe an unexpected method based on sealing of the water surface by an immiscible hydrocarbon-based liquid, such as oils, pure alkanes, and pure primary alcohols. This method, as we demonstrate through a series of experiments, enables stable supercooling of large volumes of water for long periods at temperatures well below $-10\,°C$ by eliminating the primary ice nucleation site on the water/air interface. The supercooled water can withstand vibrational and thermal disturbances with all sealing agents, and even ultrasonic disturbance if it is sealed by alcohols. In addition, we utilize this DSC approach to preserve human red blood cells (hRBCs) for as long as 100 days.

## Results

**Water deep supercooling via surface sealing.** When water molecules aggregate on the water surface (water/air interface) to form an ice nucleus, they need to overcome an energy barrier $\gamma^{ia} - \gamma^{wa}$ ($\gamma$: interfacial tension, symbols i, w, and a refer to ice, water, oil, and air, respectively) per unit area as the ice/air interface replaces original water/air interface. In comparison, the energy barrier for homogeneous ice nucleation within bulk water is proportional to the water/ice interfacial tension, $\gamma^{wi}$. This interfacial tension can be expressed via the Young's equation as $\gamma^{wi} = \gamma^{ia} - \gamma^{wa} \cos\theta_{iwa} \geq \gamma^{ia} - \gamma^{wa}$ ($\theta_{iwa}$: water contact angle on ice/water/air interface, Supplementary Note 1 and Supplementary Figure 1a). This inequality indicates that heterogeneous ice nucleation on the surface is thermodynamically more favorable than homogeneous nucleation in bulk as complete wetting ($\theta_{iwa} = 0°$) is not generally observed[27], and a receding contact angle of 12° has been reported[31]. Therefore, if the water surface is sealed by an oil phase, the energy barrier of ice nucleation at the water-oil interface would be $\gamma^{io} - \gamma^{wo}$ (symbol o refers to oil phase). Similarly, the homogenous nucleation energy barrier can be now expressed in terms of another triple interface, namely the oil/water/ice as $\gamma^{wi} = \gamma^{io} - \gamma^{wo} \cos\theta_{iwo}$, where $\theta_{iwo}$, for many oils can be nearly 0° as they are very repellent to ice (Supplementary Note 1 and Supplementary Figure 1b)[28,32]. In the case of $\theta_{iwo} \cong 0$, the energy barrier approaches the limiting case $\gamma^{io} - \gamma^{wo} \cong \gamma^{wi}$. This analysis indicates that the energy barrier of heterogeneous crystallization at the surface is elevated almost to the level of homogeneous one when the water/air interface is replaced by an oil/water interface. Accordingly, we hypothesized that surface sealing of water with an appropriate oil phase could suppress primary heterogeneous ice nucleation at the surface and enable extended storage of deeply supercooled water.

**Water deep supercooling via surface sealing with oils.** First, we cooled a large ensemble of polystyrene tubes containing 1 ml of ultra-pure water to $-13\,°C$ (Fig. 1a, b). This resulted in >90% of samples to be frozen after 24 h and nearly all samples to be frozen after 5 days. In contrast, the ultra-pure water samples could be kept in the liquid phase for a week, at the same temperature, if their surfaces were sealed by various types of immiscible oils, such as light mineral oil (MO), olive oil (OO), heavy paraffin oil (PO), and nutmeg oil (NO). Interestingly, the curdling of OO during DSC does not trigger water freezing, though the cumulative freezing frequency ($f_f$, $f_f = 1 - f_s$) increases significantly compared to water sealed by other oils (Fig. 1a). In supplementary experiments, we observed that the water degassed by vacuuming for 24 h, has similar $f_f$ as normal water, with or without oil sealing (Supplementary Figure 2). These experiments indicate that air

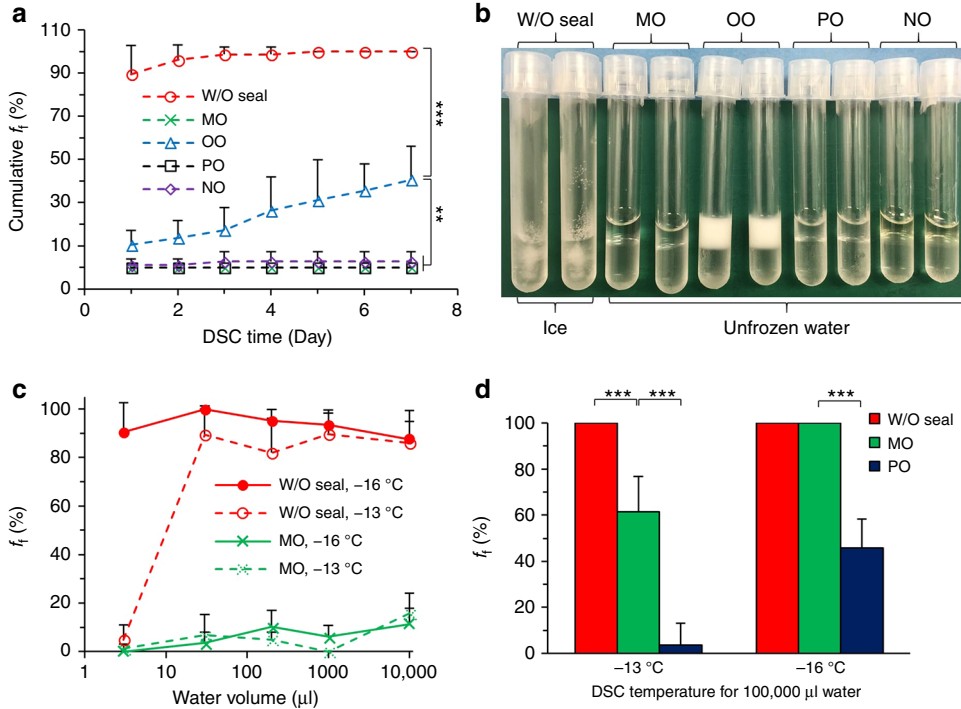

**Fig. 1** Deep supercooling of pure water enabled by surface sealing with oils. **a** Cumulative freezing frequency ($f_f$) for 1 ml water at −13 °C over 7-day deep supercooling (DSC), without sealing (W/O seal), with surface sealing by light mineral oil (MO), olive oil (OO), heavy paraffin oil (PO), and nutmeg oil (NO). Number of independent experiments $n = 6$, number of total tested samples for each case $N = 56$. NS: $p > 0.05$; *$0.005 < p < 0.05$; **$1.0 \times 10^{-6} < p < 0.005$, ***$p < 1.0 \times 10^{-6}$. **b** Corresponding samples of (**a**) post 1-day storage. **c** $f_f$ of DSC water of various volumes post 1-day storage at −13 and −16 °C. $n = 7$, $N = 272, 145, 336, 123$, and 125 for 3, 30, 200, 1000, and 10000 μl water, respectively. **d** $f_f$ of 100,000 μl water with different sealing oils and temperatures post 1-day storage, $n = 7$, $N = 35$. Error bars represent standard deviations

dissolved in the water does not play a major role in ice nucleation in our experiments. Given this result and the consistent efficacy of surface sealing by different oils on freezing reduction, we infer that the air-water interface is the primary nucleation site.

We also examined the influence of water volume on the efficacy of oil sealing for freezing inhibition. We studied the two most promising oils, MO and PO, at −13 and −16 °C for ultra-pure water ranging from $10^0$–$10^5$ μl (Fig. 1c, d, Supplementary Figure 3). We found that MO sealing can effectively suppress water freezing for water volumes up to $10^4$ μl at −13 and −16 °C. PO sealing was even more effective with a low $f_f$ throughout the entire volume range at −13 °C, and only 45.8% of samples frozen at −16 °C for the $10^5$ μl samples. In addition, 8 out of 35 (22.8%) samples of $10^5$ μl water were kept in the supercooled state at −16 °C for 100 days without any freezing event after Day-3 (Supplementary Figure 3b). While further investigations might be necessary, these observations are incompatible with conventional stochastic freezing processes (Supplementary Note 1), which implies exponential decrease of $f_s$ with time[21,22]. Alternatively, the freezing of DSC water sealed by oil could be depicted as "case-specific" that some of sealed water samples are more susceptible to crystallization than others. A reconciliation of these two cases might lie in the fact that those samples that do not freeze within our observation period have much fewer impurities and thus a much smaller exponential for the decay of $f_s$ than those that freeze within 3 days.

In order to further support our hypothesis that the water/air interface plays a dominant role in ice nucleation and subsequent freezing, we measured water freezing frequencies under differential degrees of surface sealing by MO, ranging from (I) unsealed (0 oil), (II) ring sealed along the contact line between water and tube wall (0.01 ml), (III) partially sealed with partial exposure to

air (0.1 ml), (IV) critically sealed with water surface just completely covered (0.15 ml), (V) normally sealed (0.5 ml), and (VI) over sealed with excessive oil mounted on water surface (3.5 ml) (Fig. 2a, b and Supplementary Figure 4). The results indicate that the capacity of freezing inhibition increases with the degree of sealing, with a statistically maximum plateau achieved by critical sealing (Fig. 2a). Ring sealing (0.01 ml) that nullifies the triple solid/water/air contact line has a mild effect on freezing inhibition at high temperatures (−10 °C) but is not effective below −13 °C. Taken together with partial sealing results (0.1 ml), this result implies that the contact nucleation at the air/water/solid triple interface is not as dominant as that at water/air interface especially at low temperatures. Considering the crystallization efficiency depends on the integration of nucleation probability $J$ and nucleation length (or area), the triple contact line of short length would provide smaller crystallization efficiency than the air/water interface even though it has higher $J$[33,34]. Overall, we confirmed that the water/air interface is the primary ice nucleation site for DSC water, and surface oil sealing that removes the water/air interface can effectively inhibit ice nucleation and water freezing.

We also observed that oil addition beyond the critical sealing has a statistically negligible effect on freezing suppression. This indicates that additional pressure and dampening effects, associated with a long-column of viscous oil phase, have a negligible effect on freezing inhibition. In order to further test this, we examined the effects of viscosity of the sealing agents where we used hydroxy (-OH) terminated polydimethylsiloxane (PDMS) of different chain lengths (Fig. 2c). In a similar fashion, we did not observe statistically significant differences in the capacity of freezing inhibition of PDMS with a viscosity range of 1–5 × $10^5$ cP, with the exception of 3500 cP PDMS that has almost

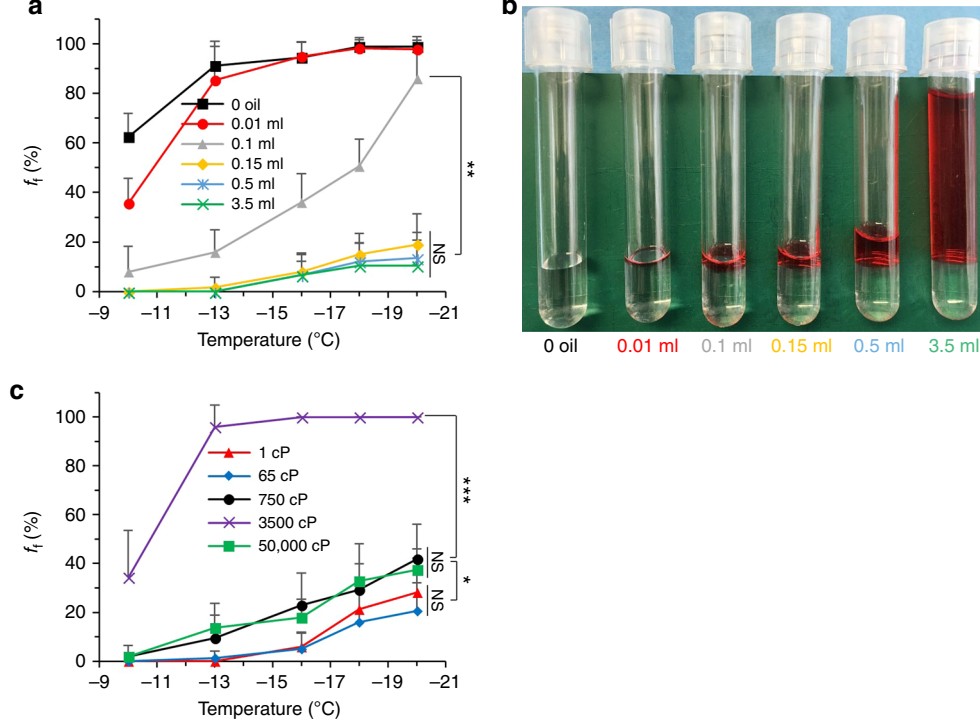

**Fig. 2** Dependence of water freezing efficiency on the volume and viscosity of sealing agents. **a** Effect of sealing oil (MO) volume on $f_f$ post 1-day DSC at different temperatures. $n = 6$, $N = 70$. **b** Side view of corresponding samples of **a**. MO includes Oil Red O for staining and imaging. 0 oil, 0.01 ml, 0.1 ml, 0.15 ml, 0.5 ml, and 3.5 ml indicate no seal, ring seal, partial seal, critical seal (just complete surface seal), standard seal, and over seal by MO, respectively. **c** Effect of viscosity of sealing agents on $f_f$ post 1-day DSC at −16 °C. The sealing agents are hydroxy (OH) terminated polydimethylsiloxane (PDMS) of different chain lengths and viscosities. $n = 5$, $N = 56$. Error bars represent standard deviations

no freezing suppression effect. We hypothesize that this odd behavior is likely due to the formation of an ordered structure between water and this particular PDMS on the interface through hydrogen bonding, which closely matches the lattice of hexagonal ice[35].

**Water deep supercooling via surface sealing with alkanes and alcohols**. Most oils are complex mixtures of alkanes, saturated cyclic alkanes, alkylated aromatic groups, and fatty acids among other hydrocarbon compounds. In an effort to more systematically study the observed freezing inhibition effect of supercooled water sealed with an immiscible hydrocarbon phase, we studied two prototypical families of hydrocarbons: linear alkanes and their corresponding primary alcohols of different lengths (Fig. 3). Specifically, we have studied alkanes ($C_mH_{2m+2}$, denoted $C_m$, $m = 5 – 11$) and primary alcohols ($C_mH_{2m+1}OH$, denoted $C_mOH$, $m = 4 – 8$) as the sealing agents for DSC water at −20 °C. Since linear alkanes have very low polarity, they have weak interaction with polar water molecules. On the other hand, the primary alcohols, which are amphipathic, can form strong hydrogen bonds with water through their hydroxyl group (hydrophilic end) and even stable ordered interfacial structures. However, a binary combination of an alkane and an alcohol should not be utilized as a sealing agent as it forms cooperative hydrogen bonding, heterogenous microdomains, and interfaces[36,37], pumping water into the mixture to produce milky emulsions on top of water (Supplementary Figure 5).

We found that $f_f$ of DSC water, at −20 °C, sealed with alkanes decreases monotonically with increasing carbon number $m$ and chain length $l$ (Fig. 3a). The capacity of alkanes in freezing

inhibition matches that of MO (Fig. 2a at −20 °C) at $m > 9$. This coincides with the observation that mineral oils tend to have hydrocarbons with alkane chain lengths above 10. While oils comprise of many different hydrocarbons, alkanes make up a major fraction of their composition. Accordingly, the trend of higher freezing inhibition with longer alkane chain lengths, might also partially explain the differences in $f_f$ between PO and MO (Fig. 1d) among other effects from other hydrocarbon groups that we have not yet studied. PO likely consists of longer carbon chain alkanes than MO based on their densities (PO ~ 0.855 – 0.88 vs MO ~ 0.838 g ml^−1) and dynamic viscosities (PO ~34 vs MO ~23 cP[38]). On a molecular level, the mechanism for this trend might lie in the structure of the alkane/water interface. It has been observed that an interfacial electron depletion layer with a thickness $\delta$ exists between water and hydrophobic alkane chains by both X-ray reflectivity (XR) measurements[39–41] and atomistic molecular dynamics (MD) simulations[42,43]. The few water molecules in the depletion layer (electron density < 40% that of bulk water[44]) can buckle in the intermolecular space near the ends of alkane molecules (Fig. 3b), and create a template for the formation of an ice nucleus[29]. The alkane chains adjacent to the water molecules preferentially have their longest axis parallel to the water interface with a tilt angle $\beta$[39]. This tilt angle increases with $m$ and $l$, resulting in a more parallel orientation for longer alkanes[39]. Accordingly, longer alkane chains are expected to reduce the corrugation and roughness of the interface on the side of alkanes. This, consequently, would decrease the number of buckled water molecules and nucleus templates, and thus lower the probability of heterogeneous ice nucleation on that layer[29]. These expectations are in line with our observations of decreasing freezing frequencies for longer alkane chain lengths. From the perspective

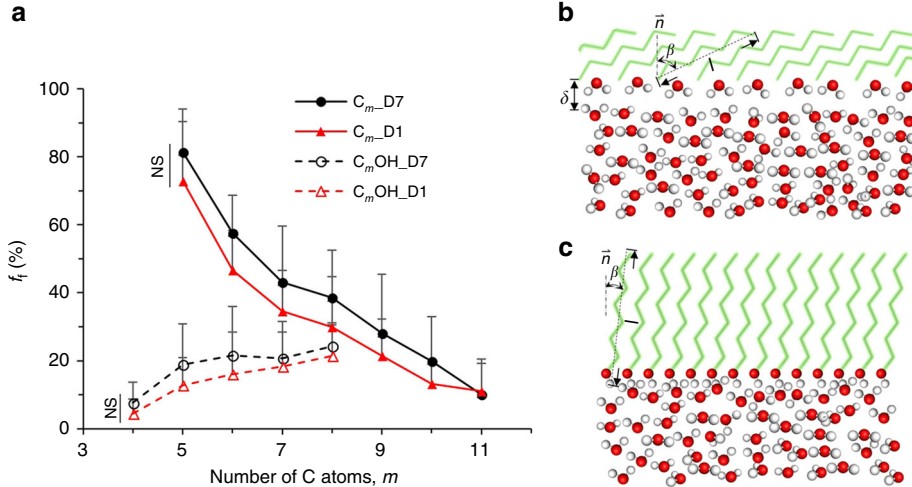

**Fig. 3** Deep-supercooled water sealed with linear alkanes and primary alcohols. **a** $f_f$ of 1 ml deep-supercooled (DSC) water at −20 °C. $n = 7$, $N = 87$. Error bars represent standard deviations. When $m > 11$ for linear alkanes and $m > 8$ for primary alcohols, the sealing agents are frozen at −20 °C and cause DSC water frozen. When $m < 5$, the linear alkanes are gaseous under atmospheric condition and not suitable for sealing. When $m < 4$, the primary alcohols are miscible with water and not suitable for sealing either. **b**, **c** Schematic configurations of alkane/water (**b**) and alcohol/water interface (**c**), respectively. The alkane and alcohol molecules are displayed without aliphatic hydrogen atoms and colored in light green. The O and H atoms in hydroxyl group of alcohol and water are shown in red and white dots, respectively

of thermodynamics, longer and flatter-oriented alkanes results in fewer and sparser buckled water molecules in the interface serving as nucleation template, which implies smaller contact region between ice embryo and sealing alkanes, smaller $\theta_{iwo}$ (even though they are already much smaller than $\theta_{iwa}$), and higher energy barrier for heterogeneous ice nucleation (Supplementary Figure 1b).

On the other hand, $f_f$ of DSC water, at −20 °C, sealed with alcohols increases with $m$ and $l$. For example, $f_f$ equals 4.4% and 21.4% after 1-day DSC for $C_4OH$ and $C_8OH$, respectively. $C_4OH$ has a higher freezing inhibition capacity compared to $C_5OH$ ($f_f =$ 4.4% for $C_4OH$, $f_f =$12.8% for $C_5OH$ sealing). Nevertheless, $C_4OH$ has a small but relatively higher solubility in water than $C_5OH$, and accordingly $C_5OH$ might be the optimal choice for sealing. The different behavior of alcohols with respect to chain length might be due to the different structures of the alcohol/ water interface compared to that of alkane/water interface (Fig. 3c). Unlike the alkanes which prefer a parallel orientation, the primary alcohols orient perpendicularly to the interface with a small $\beta$ (usually less than 30°)[45,46]. The primary alcohols align their hydroxyl (-OH) heads toward the interface to form hydrogen bonds with water molecules. Accordingly, no depletion layer of interfacial water exists as in the alkane/water interface. The 2D layer of interfacial water molecules are strongly hydrogen-bonded to the hydroxyl groups, with their H atoms pointing toward alcohol as revealed by heterodyne-detected vibrational sum frequency spectroscopy (SFG)[47]. Therefore, structures and dimensions of the contacting layer of amphilic alcohols essentially determine the distribution and arrangement of interfacial water molecules, and the formation of heterogeneous ice nucleus[35,48].

Experimental measurements via grazing incidence X-ray diffraction (GIXD) and MD simulation of ice nucleation in droplets under monolayers of long primary alcohol chains with $16 \leq m \leq 31$, revealed a very low tilt angle $\beta$ (~7.5 – 12°) and a very good lattice match between hexagonal ice and the alcohol structure for $29 \leq m \leq 31$[45,46,49], resulting $T_f$ as high as −1 °C for these longest chains. As $m$ and $l$ decrease, $\beta$ increases up to ~19° for $m = 16$[46]. In conjunction, a greater lattice mismatch between hexagonal ice lattice and ordered alcohol layer at the interface

along with a lower ice nucleation efficiency and $T_f$ were observed[35,45,46,49]. For shorter alcohols ($4 \leq m \leq 8$) in this study, larger tilt angles would ensue as evidenced by $\beta = 28°$ for $m = 6$ and $\beta = 30°$ for $m = 5$[46], causing greater lattice mismatches between hexagonal ice and ordered alcohol structure given the general structural similarity of primary alcohols. Compared to longer alcohol chains, the interfacial -OH groups anchored to smaller alcohols have stronger in- and out-of-plane fluctuations at the same temperature. We, therefore, expect that lattice mismatch and the –OH group fluctuations can destabilize any ordered domain of crystalline water and impede the formation of ice nucleus of critical size[49]. Given that both effects are larger with smaller chain lengths, we expect that higher nucleation inhibition can be achieved by smaller primary alcohols, in line with our experimental observations. From the perspective of thermodynamics, greater lattice mismatch and interface fluctuation associated with shorter alcohol molecules directly reduce the probability of the formation of icing template of critical size for successful nucleation, which indicates smaller stable contact area between ice nucleus and sealing alcohols and thus, higher free energy barrier for heterogeneous ice nucleation at the interface. Once again, we observed that there is no significant difference of $f_f$ between 1-day and 7-day storage when sealed by either alkanes or alcohols. This further suggests the case-specific, rather than stochastic, nature of water freezing with oil sealing that we have previously discussed.

**Stability tests for deep-supercooled water.** Having established the efficacy of the DSC approach using either oils or pure alkane and alcohol phases, we then studied its stability under vibrational, thermal, and ultrasonic disturbances. Vibrational disturbances were introduced by placing DSC water onto a shaking plate with various shaking speeds and frequencies. When the DSC water (−20°) is sealed by MO, its $f_f$ is 0% and 5.6%, respectively, under 0.84 g and 2.1 g centrifugal acceleration (Fig. 4a), which are much higher than ac/deceleration forces of a commercial airliner (0.2 – 0.4 g) during potential transporation. Thermal disturbances were induced by putting the DSC samples into 37 °C incubator or plunging them into 37 °C

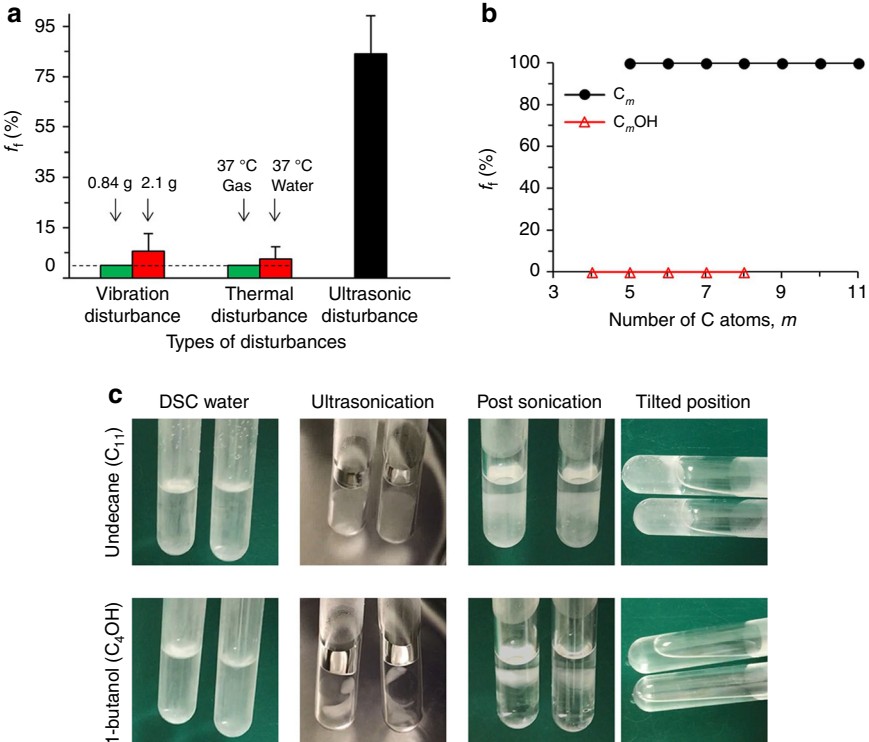

**Fig. 4** Stability tests for 1 ml deep-supercooled water at −20 °C. **a** $f_f$ of deep-supercooled (DSC) water sealed by MO under various disturbances. Vibrational disturbance was imposed by shaking plate with different shaking frequencies and centrifugal forces (i.e., 0.84 g or 2.1 g). Thermal disturbance was imposed by placing or plunging the DSC tubes into 37 °C incubator (37 °C gas) or water bath (37 °C water). Ultrasonic disturbance was introduced by putting the DSC tubes into 40 kHz ultrasonic water bath. $n = 6$, $N = 48$. Error bars represent standard deviations. **b** $f_f$ of DSC water sealed by linear alkanes and primary alcohols under 40 kHz ultrasonic disturbance. $n = 3$, $N = 24$ (except for $C_5$, $N = 8$). **c** Representative image sequences of ultrasonication tests for DSC water sealed by linear alkanes or primary alcohols

water bath with warming rate of $10^0$ C min⁻¹ (heated by natural convection in air) or $10^2$ C min⁻¹ (heated by forced convection in water), respectively. Very few (0% for gas warming, 2.5% for water warming) of the samples freeze under these thermal fluctuations. In contrast, these samples cannot endure ultrasonication in 40 kHz ultrasonic water bath (Fig. 4a and Supplementary Movie 1), with $f_f$ of ~ 84%. This is probably due to the vigorous collapse of cavitation bubbles in water during ultrasonication[18], which would cause ultrahigh local pressure (>1 GPa)[50], and therefore, significantly increase equilibrium temperature $T_e$ and the degree of supercooling $\Delta T$.

Upon the instability of DSC water sealed by MO under ultrasonication, we further tested its stability sealed by pure alkanes and primary alcohols. DSC water sealed by alkanes freeze immediately upon being ultrasonicated (Fig. 4b–c and Supplementary Movie 2), which is consistent with previous observation of MO sealed water since MO has a high content of various alkanes. On the contrary, none of the samples freeze upon ultrasonification if they were sealed by any of the primary alcohols (Fig. 4b). Instead, the sealing alcohols would be emulsified with supercooled water, starting from the interface and then evolving toward supercooled water (Fig. 4c and Supplementary Movie 3). The exact mechanism of the freezing resistance of DSC water sealed by alcohols to ultrasonic disturbance is still unknown, and one hypothesis would be that ultrasound preferentially transduces its energy into joint molecular motion at interface due to the hydrogen bonding between water and amphilic alcohols to form nanoemulsion[51], rather than cavitation bubbles for ice nucleation in DSC water.

**Deep supercooling of hRBCs for extended preservation**. In addition to DSC of pure water, we have utilized the surface sealing method to achieve DSC for aqueous solutions and cell suspensions to demonstrate extended supercooling preservation of biological specimens (Fig. 5). The current clinical standard for preservation of hRBCs is via conventional cold storage at 4 °C with the CP2D + AS-3 (anticoagulant citrate phosphate double dextrose supplemented with additive solution 3) solution[52]. This standard approach for hRBC preservation can provide storage for a maximum of 42 days[53], beyond which the cells experience irreversible storage lesions including hemolysis as shown in Fig. 5a. According to the Arrhenius relationship, preservation at deep subzero temperatures would slow down cellular metabolism and decay rates, extending this biopreservation period. Accordingly, we have preserved hRBCs at as low as −16 °C for an extended period of 100 days. Specifically, using surface sealing by PO, we successfully supercooled 1 ml suspensions of hRBCs in either conventional cold storage solution, CP2D + AS−3[52], or University of Wisconsin solution supplemented with 5% (w/v) trehalose (UW + Tre), for as long as 100 days at −7, −10, −13, and even −16 °C (see Supplementary Figure 6 for $f_f$). The suspension volume could be extended to the magnitudes of $10^1$ even $10^2$ ml as 5 out of 6 vials of 30 ml hRBC suspension (500 million cells) kept unfrozen over 365-day supercooling at −13 °C in our pilot experiments. In CP2D + AS-3 solution, hRBCs experience remarkable hemolysis as shown by the presence of dark spots (RBC debris without holding hemoglobin) in the micrographs of Fig. 5a and reduced recovery rates of hemoglobin in Fig. 5b, c, especially at deep subzero temperatures. Surviving hRBCs become spherically shaped, losing the distinctive biconcave disc form of

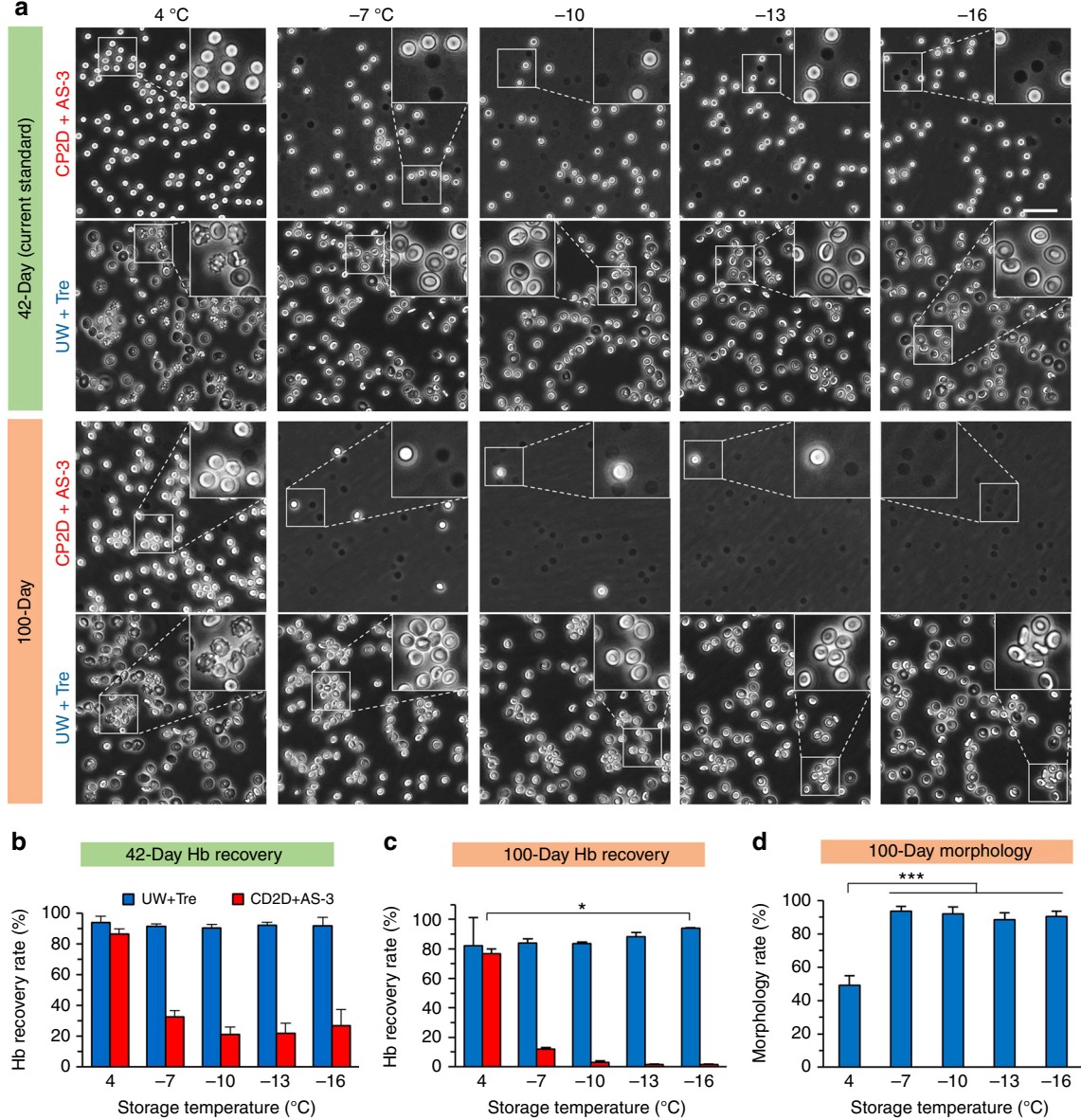

**Fig. 5** Cold storage and deep-supercooled preservation of human red blood cells. **a** Preservation of human red blood cells (hRBCs) for 42 and 100 days at various temperatures in conventional anticoagulant citrate phosphate double dextrose (CP2D) supplemented with additive solution 3 (AS-3) (CP2D + AS-3), and UW solution supplemented with 5% (w/v) trehalose (UW + Tre). Black dots are the debris of lysed RBCs without holding hemoglobin. Successful DSC for 1 ml suspension of 10 million hRBCs in polystyrene tubes was achieved by surface sealing with 0.5 ml PO. Scale bar is 50 μm. **b** Hemoglobin recovery rate of RBCs post 42-day storage. $n = 3$. **c** Hemoglobin recovery rate of RBCs post 100-day storage. $n = 3$. **d** The percentages of RBCs with normal morphology and size post 100-day storage in UW + Tre solution. Bright disk-shaped RBCs with smooth plasma membrane and diameter 5 μm ≤ D ≤ 10 μm are regarded as normal morphology. $n = 3$. The total number of counted cells $N = 1800 ± 50$ for each case. Error bars represent standard deviations

fresh hRBCs (Supplementary Figure 7). However, in UW + Tre solution, hRBCs do not undergo noticeable hemolysis, and the recovery rates of hemoglobin at −16 °C (94%) is higher than that of conventional storage at 4 °C in CP2D + AS-3 solution post 100-day storage (76%, Fig. 5c). In addition, they can also maintain their intact, shiny, and discoid-shaped phenotype at DSC temperatures (the second and fourth rows of Fig. 5a), resulting in higher percentages of normal morphology (>88%, exclusion of serrated, spherical, swollen, or shrunk cells) than cold storage at 4 °C (49%, Fig. 5d). These results demonstrate that DSC via surface sealing can effectively prolong the storage time of hRBCs to 100 days combined with the optimization of preservation solutions.

## Discussion
In the preceding, we demonstrated a seemingly counterintuitive approach to achieve long-term DSC of large volume water by using a hydrocarbon-based immiscible phase to seal the water surface. Our initial observations with laboratory grade oils demonstrated that replacing the water/air interface, which is the primary ice nucleation site, with a water/oil interface dramatically inhibits stochastic freezing processes. The seemingly time independent nature of the freezing frequency of oil-sealed water suggests that its freezing might be case-specific rather than stochastic. Our studies with linear alkanes and primary alcohols suggest that freezing inhibition can be achieved by surface sealing with starkly different interfacial structures and

microscopic mechanisms, which results in different trends of inhibition capacity correlated to the chain length. While all sealed DSC water show great stability under vibrational and thermal disturbances simulating normal storage and transportation conditions, only the primary alcohol sealed supercooled water can withstand ultrasonication. While we have hypothesized about possible macroscopic (thermodynamic) and microscopic mechanisms that might explain our observations, further studies are warranted to test, confirm, and improve upon them. In addition, electric field on interfacial water introduced by surface sealing by alkanes and alcohols also could contribute to the inhibition or promotion of ice nucleation[54]. But the direction of electrical field ("positive" or "negative") and induced orientation of water molecules ("hydrogen down" or "hydrogen up") are not consistent even contradictory according to SFG measurements and MD simulations[40,47,55], whose effects on water freezing require further investigation. Especially, since most existing literature focuses on longer alkane and alcohol chains at the water interfaces and the resulting molecular structures, computational and experimental studies with short chains might prove useful. Similarly, careful measurements of interfacial properties and structures at low temperatures with mixed and pure hydrocarbons can shed further light on why some oils are more effective than others.

Given that monolayers of primary linear alcohols of long chains ($m \geq 16$) have been historically used to initiate ice nucleation, our results with the short chains to prevent nucleation expand the use of alcohols to provide a robust control mechanism over the temperature at which nucleation can be achieved in an aqueous solution. Further studies with different families of hydrocarbons and their mixtures will be aimed to expand this robust control of supercooling to enable various applications. Beyond its fundamental implications, DSC of large volumes of aqueous solutions can enable previously prohibitive applications, and provide unprecedented biopreservation methodologies for cell, tissue, and organ engineering and transplantation, as well as other areas, such as food preservation. Given our prior experience and interest in both organ and cell preservation using supercooling and the limitations we have previously encountered in terms of temperatures, volumes and durations for preservation, we believe that the DSC via the surface sealing with immiscible phases will be vital in advancing these applications forward. The immediate goal is to translate this approach to preservation of other types of cells that are amenable to DSC preservation as we demonstrated for hRBCs here, and then translate such results to the clinic. We will then explore tissue and organ preservation with DSC approaches.

## Methods

**Experimental materials.** For all experiments in this study, DNase/RNase-free distilled water (Life Technologies/Thermo Fisher Scientific, USA) was used to minimize potential pollutants or ice-nucleating agents, except DSC trials of 100 ml water where deionized (DI) water (resistivity $R = 18.2$ MΩ) produced by a deionizing water system (METTLER TOLEDO Thornton, USA) was used. All water containers (dishes, 96-well plates, round-bottomed tubes, and bottles, Corning, USA) were made of polystyrene, and clean and sterile before experiments. All oil phases, such as light mineral oil (MO), heavy paraffin oil (PO), olive oil (OO), nutmeg oil (NO), alkanes, and alcohols, used for water surface sealing were purchased from Sigma-Aldrich, USA, and their purities were at least 99%.

**Water supercooling procedures.** The loading of water into containers was performed in a chemical hood to avoid contamination of the samples by pollutants or dust particles in the air. Water of small volume (< 1 ml) was loaded into containers (dishes or 96-well plates) using clean and sterile tips (Thermo Fisher Scientific, USA) and calibrated pipets (PIPETMAN, Gilson, USA), while that of large volume (≥ 1 ml) was loaded into containers (round-bottomed tubes or bottles) using serological pipets (Thermo Fisher Scientific, USA) by pipette filler (Drummond Scientific, USA). We note that as water droplets smaller than $10^0$ μl are subject to significant evaporation during long-term DSC experiments, and those bigger than

$10^5$ μl (100 ml) beyond the volume capacity of the freezing chamber, they were not investigated in this study. After loading water samples into the containers, oil phase was gently added onto the water surface using serological pipets, trickling down along the wall of containers to avoid splashing or trapping air bubbles at the interface. The water-laden containers (with or without sealing oil) were transferred into portable temperature-controlled freezers (Engel MHD-13, Engel, USA) that were placed in 4 °C cold room to minimize temperature fluctuations, or stored in −20 °C freezer (Thermo Fisher Scientific, USA). The temperatures within these freezers were verified by Toluene-filled low-temperature thermometer (Sigma, USA).

**Water degassing for supercooling tests.** To examine the effects of dissolved air in water on ice nucleation and water freezing, the water was vacuumed at a pressure below $10^{-4}$ atmosphere for 24 h to extract dissolved air molecules. The degassed water was, then, gently pipetted into tubes and sealed with mineral oil (MO) for supercooling tests at −16 °C. The air content of the degassed water is significantly lower than that of normal water without degassing, as no air bubbles emerge from the degassed water (second row of Supplementary Figure 2b) under vacuum. The same procedure was carried out for normal water for comparative purposes, and several big air bubbles can be observed after 3-hour degassing (first row of Supplementary Figure 2b).

**Stability tests.** To test the stability of DSC water sealed by oil phase, three types of disturbances, vibrational, thermal, and ultrasonic disturbances, were studied. For vibrational disturbance test, DSC tubes were placed on shaking plate (Labline 4625 titer shaker, Marshall Scientific, USA) with shaking speed 500 and 800 rpm for 30 s, which give rise to the centrifugal acceleration of 0.84 and 2.1 g (g is gravitational acceleration), respectively. To prevent heat transfer, the tubes were wrapped with thick tissue paper in tube racks, all of which had been previously cooled to −20 °C in freezer. The temperature of the DSC water would not change noticeably during experiments given the brief shaking period and thick insulation layer. For thermal disturbance test, the DSC tubes were put into 37 °C incubator (warmed by air) or plunged into 37 °C water bath (warmed by water). Therefore, DSC water would experience different warming rates and temperature gradient. For ultrasonic disturbance test, DSC tubes would be plunged into 4 °C ultrasonic water bath. The sonicator (Branson B-200, TMC Industries, USA) generates 40 kHz ultrasonic wave with power 30 W. The freezing of static DSC water can be determined by visual inspection for the change of sample transparency (from transparent to opaque, Fig. 1b), while for DSC water in stability tests, the occurrence of freezing can also be determined by tube tilt as frozen water cannot flow freely (Fig. 4c).

**Supercooling preservation for hRBCs.** Fresh hRBCs were purchased from Zen-Bio (NC, USA) and utilized immediately upon arrival. They were suspended in either anticoagulant citrate phosphate double dextrose (Haemonetics, MA, USA) supplemented with additive solution 3 (Haemonetics, MA, USA) (CP2D + AS-3), or University of Wisconsin solution (Bridge to Life, SC, USA) supplemented with 5% (w/v) trehalose (Sigma-Aldrich, USA) (UW + Tre), with a concentration of 10 million/1 ml. Next, 1 ml hRBC suspension was transferred into 5 ml round-bottomed polystyrene tube, sealed by 0.5 ml paraffin oil (PO), and stored in portable temperature-controlled freezer for desired periods. Post certain storage time (42 or 100 days), samples were warmed up in 37 °C incubator, followed by the removal of sealing oil via aspiration. Then hRBCs were resuspended into phosphate-buffered saline (PBS) for the measurements of hemoglobin recovery rate and examination of cell morphologies. To release the hemoglobin in intact hRBCs, deionized (DI) water was added in RBCs to burst cells via hypotonic osmolarity, followed by vigorous vortex for 3 min. The concentration of hemoglobin in solution was determined by direct spectrophotometry, utilizing formulas employing absorbance at 415, 380, and 450 nm ("Allen correction" formula[56,57]) i.e., $C_{he}$ (mg L^−1) = $1.68A_{415} - 0.84A_{380} - 0.84A_{450}$, where Che is the concentration of hemoglobin, $A_{415}$, $A_{380}$, and $A_{450}$ are the absorbance values at 415, 380, and 450 nm, respectively. The recovery rate of hemoglobin in hRBCs for each sample was calculated by $\eta = C_{he,pe}/(C_{he,pe} + C_{he,su}) \times 100\%$, where $\eta$ is the recovery ratio of hemoglobin in RBCs, $C_{he,pe}$, and $C_{he,su}$ are hemoglobin concentrations of lysed RBCs in the pellet (hemoglobin is not released by hemolysis during storage or processing) and supernatant solution of RBCs (hemoglobin released during storage or processing). In addition, hRBCs suspended in PBS were observed and imaged by phase-field microscopy. To obtain the rate of normal morphology (bright disk-shaped cells with smooth membrane rather than serrated, spherical, swollen, or shrunk cells) in preserved hRBCs, around 600 cells (total 1800, $n = 3$) were counted and the rate of normal morphology was calculated.

**Data analysis.** All data were organized and reported as the mean ± standard deviation from at least three independent runs of experiments ($n > 3$); further information on sample numbers are disclosed in figure captions. The statistical significance of mean values between two groups was determined by Microsoft Excel based on Student's two-tailed $t$-test, assuming equal variance. Although a $p$-value less than 0.05 is generally regarded as statistically significant, different ranges of $p$-value (NS: $0.05 < p$, *$0.005 < p < 0.05$, **$0.005 < p < 10^{-6}$, ***$p < 10^{-6}$) were provided to show different degrees of significance.

**Data availability**. All relevant data are available from the authors upon request.

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

## Acknowledgements

We are grateful to Prof. Mehmet Toner for the very helpful discussions and his suggestions. We would also like to thank the NIH for funding this work through grants no. 5P41EB002503 (BioMEMS Resource Center), 1R21EB020192, 5R01EB023812.

## Author contributions

H.H., M.L.Y. and O.B.U. conceived the project. H.H performed experiments and wrote the manuscript draft. All authors analyzed the data and revised the manuscript.

## Additional information

**Competing interests:** The authors declare no competing interests.

