## [Peer Review File · Nature Communications]

Reviewers' comments:

Reviewer #1 (Remarks to the Author):

In this manuscript, Huang et al. describe a simple method to achieve long-term deep supercooling of water and inhibition of ice formation in large volumes of water based on surface sealing of water with oil. The authors demonstrate the effectiveness and generality of this method by using common oils, pure alkanes, and short-chain alcohols and by examining a variety of experimental conditions. The study appears to have been carefully executed, the results are robust, and their explanations for the observed effects seem logically and scientifically plausible. The significance of this study lies in the elegance of a very simple science-based method to address a difficult challenge with practical implications, i.e., simultaneous achievement of low temperature, large volume, and long term stability for supercooling of water. As such, the results of this study would be of interest to the broad readership of Nature Communications and could even be appreciated by general audience with casual interest in science. Provided that some questions and suggestions for improving the manuscript (see below) are adequately addressed, I would recommend publication of this study.

Comments:

1. It is not clear from the manuscript exactly what criteria were used to determine whether the water was frozen or not. Is it simply based on tilting of the sample tubes and visual inspection? If so, provide a brief description of the visual inspection method used, in order to convince the reader that the method was sufficient to support the conclusions. If a method other than visual inspection was also used, describe that as well.
2. The difference in the supercooling behavior between pure alkanes and alcohols as surface sealants is scientifically very interesting, and their explanations seem plausible. In order to strengthen their conclusion about the role of alcohols at the oil water interface, it is recommended that the authors perform additional water supercooling experiments using a binary alkane/alcohol mixture with alcohol being a very minor component, just sufficient to form a few surfactant monolayers at the oil-water and oil-vapor interfaces. If the authors' conclusion about the role of interfacial alcohol is correct, the deep supercooling observed for pure alcohols should be achieved also by using a binary alkane/alcohol mixture with a very tiny alcohol fraction.
3. Source of common oils. In order that others can repeat the experiment, it is suggested that the authors add, in the materials and methods section of SI, the sources of common oils used in this study, i.e., PO, MO, OO, and NO.
4. In captions for Figs. 3 and 4, note the volumes of water used for the results shown.

Reviewer #2 (Remarks to the Author):

In this paper, Huang et. al. measured the cumulative freezing frequency of water (>1 ml) in presence and absence of different oils on water surface. The results show remarkable decrease of freezing frequency in presence of oils. Since the oils contain a mixture hydrocarbons and even fatty acid, the authors have examined the change in freezing frequency in presence of pure oil-like compounds, such as linear alkanes and alcohols; and the results are qualitatively similar to that of the oils. Based on these results, the authors made an attempt to provide a molecular level picture of the hindered ice nucleation at the oil-water interface. The results are interesting, and may be publishable in Nat. Comm. after consideration of the following comments:

1. In the abstract it is mentioned that, "The relationship of this capacity with the chain length, however, shows opposite trends for alcohols and alkanes due to their drastically different interfacial structures with the water molecules." Similar statements are also there in the main text.

However, if we see Figure 3A which is the basis of the above statements, the opposite behavior is not so obvious. For example, in the case of alkane, on increasing the no of carbon from 5 to 8, ff decreases by $\sim 50\%$. But in the case of alcohol, the ff increases only by $\sim 5\%$. Now given the error bar of the results, it is reasonable to believe that in case of alcohols (except butanol which is much more soluble in water than the other alcohols examined) ff has already reached its optimum value even with c5-alcohol. Interestingly, the ff values with alcohols (c5 to c8) are close to the ff value of C10 or c11 alkanes. Thus, it seems to us that the alcohols are more efficient than alkanes in inhibiting ice nucleation at interface, at least in the regime of short chains ($\sim C < 10$).

2. To explain the differences in ff values between alkane and alcohol, the authors considered different structure and orientation of the alkyl chains as well as that of the water at the interface. The reasons of different behavior of alkane and alcohol presumably lie with the different interfacial structure. However, the molecular level understanding of the alkane-water and alcohol-water interfaces are not well established yet, experimentally. In fact recently SFG studies have suggested negative charging of the aqueous interface in presence of alcohol and alkane. As a result, the molecular level explanation of the alcohol and alkane induced freezing inhibition appears to be speculative. It needs further investigation to propose such a picture.

3. Line 167- 170: Extrapolation of the alkyl chain length dependence to different kinds of oils (PO vs MO) might lead to an over analysis of the result. This because the oils are mixture of many components including fatty acids which is expected to have stronger effect on interfacial water than alkane.

4. Line 82: the homogeneous ice nucleation has been explained by the energy terms of the triple interface as a form of Young equation (Line 82 in the text) and the corresponding model is shown in Figure S1A. Is the Figure consistent with the above equation? The author should show the derivation of the equation (in Supporting information) by the vector component analysis.

Reviewer #3 (Remarks to the Author):

This work reported long-term super-cooling of a few millimeter of pure water at temperatures as low as -20 C. This was achieved by sealing water with immiscible oil or alcohol. The authors also tried to explain the observed phenomena based on theoretical analyses of interfacial tension. Although the observed phenomena is somewhat interesting, it is unclear if the phenomena could still be observed if pure water is replaced with more biologically relevant samples including saline, cell culture medium, and suspensions of cells, tissues, and organs. Therefore, the manuscript as it stands currently is more appropriate for a physics journal than Nature Communications. My major and minor concerns on this study are given below:

Major

1. The authors mentioned in multiple places in the manuscript that the reported super-cooling phenomenon might be applied for the preservation of cells, tissues, and organs. However, no data is provided to support this statement at all. This is a major concern, because the equilibrium in the super-cooled water may not be maintained long if there are impurities in the water according to the authors (lines 122-124). For preservation of cells/tissues/organs, the sample is not pure water but water with impurities (i.e., salts, proteins, cells, tissues, or organs). Therefore, without convincing data on long-term preservation of organs, tissues, or at least cells, this work is more appropriate for publication in a physics journal.

2. Lines 120-121, what are the unique and slightly different microstructures on the interface? Experimental data are needed to show the microstructures.

3. Lines 191-192, experimental data are needed to support this statement.

4. Line 208, extrapolating is not a reliable approach and experimental data are needed to support the extrapolation.

Minor

5. Lines 26-28, this is an exaggerated statement. I strongly disagree that the freezing and super-cooling of water are the least understood phenomena in our daily lives and scientific research. There is good amount of knowledge on this topic compared to many other topics including cell biology.

6. Line 49, this statement is incorrect. Only pure water freezes at 0 C under 1 atm, but it takes a long time.

7. Line 78, I do not see the symbol θ in the mathematical equation.

8. Lines 83-90, it is unclear why the contact angle for ice-water-oil should be nearly 0, since they are immiscible. In contrast, the contact angle for ice-water-air should be close to 0 because they are more miscible.

9. Lines 271-272, this statement is questionable. For example, PVA is a long chain alcohol and has been used as the ice nucleation inhibitor instead of initiating ice nucleation.

Response Letter

We thank you and all three reviewers for the insightful comments, critical questions, and very helpful suggestions. We believe that we have now tended to all these outstanding issues including the addition a) of new data on the interesting experiment of binary mixtures of alkanes/alcohols proposed by reviewer #1, b) a clearer hypothetical picture of the interfacial microscopic structures and vector component analysis of interfacial tensions suggested by reviewer #2, and c) data on preservation of human Red Blood Cells (hRBCs) for 100 days under DSC conditions (at -16, -13, -10, and -7 °C) suggested by Reviewer #3 and yourself, among others that we detail below. The new data on hRBCs preservation, despite its preliminary nature, illustrates the feasibility of translating the DSC approach to storage of biological specimens among many other possible novel applications. We believe that through the helpful comments of the reviewers and this revision, our manuscript has become richer, more detailed, and clearer.

In the following point-by-point response we first introduce the question/suggestion/comment of each reviewer in black non-bolded text and then present our response in blue bolded text.

We are deeply grateful for the effort and attention to detail by all three reviewers and we welcome additional feedback should they have any further questions.

Best Regards,

Dr. Berk Usta

PS: We include two files for both the main text and the supplemental information file a) one which marks all the changes made to the text and b) one which shows the final version of the main text so that reviewers can easily identify the changes. The line numbers we cite below correspond to those in the marked file.

Reviewer #1:

In this manuscript, Huang et al. describe a simple method to achieve long-term deep supercooling of water and inhibition of ice formation in large volumes of water based on surface sealing of water with oil. The authors demonstrate the effectiveness and generality of this method by using common oils, pure alkanes, and short-chain alcohols and by examining a variety of experimental conditions. The study appears to have been carefully executed, the results are robust, and their explanations for the observed effects seem logically and scientifically plausible. The significance of this study lies in the elegance of a very simple science-based method to address a difficult challenge with practical implications, i.e., simultaneous achievement of low temperature, large volume, and long-term stability for supercooling of water. As such, the results of this study would

be of interest to the broad readership of Nature Communications and could even be appreciated by general audience with casual interest in science. Provided that some questions and suggestions for improving the manuscript (see below) are adequately addressed, I would recommend publication of this study.

Comments:

1. It is not clear from the manuscript exactly what criteria were used to determine whether the water was frozen or not. Is it simply based on tilting of the sample tubes and visual inspection? If so, provide a brief description of the visual inspection method used, in order to convince the reader that the method was sufficient to support the conclusions. If a method other than visual inspection was also used, describe that as well.

Response: We apologize for this oversight and omission. We utilized a direct visual inspection (for all samples) approach and additional inspection of flow/deformation upon tube tilting (for samples used for testing stability) to determine the freezing events of large-volume water (≥ 1 ml). The large-volume frozen water in tubes is not only much more opaque than supercooled water, but also has a nonuniform transparency due to the heterogeneity of crystals across the bulk of the ice. (Fig. 1(B)). Accordingly, the visual inspection provides a convenient and convincing way to determine water freezing. In addition, sample tubes were tilted during the stability tests to confirm water freezing events (Fig.4(C)) since frozen water cannot flow under tilting conditions. We have incorporated a writeup on these inspection methods in our revision in the “Materials and methods” section in Supplemental Information (SI) file (Lines 48 - 51) to clarify the visual inspection and tube tilt methods to determine water freezing.

2. The difference in the supercooling behavior between pure alkanes and alcohols as surface sealants is scientifically very interesting, and their explanations seem plausible. In order to strengthen their conclusion about the role of alcohols at the oil water interface, it is recommended that the authors perform additional water supercooling experiments using a binary alkane/alcohol mixture with alcohol being a very minor component, just sufficient to form a few surfactant monolayers at the oil-water and oil-vapor interfaces. If the authors' conclusion about the role of interfacial alcohol is correct, the deep supercooling observed for pure alcohols should be achieved also by using a binary alkane/alcohol mixture with a very tiny alcohol fraction.

Response: We thank you very much for this insightful and interesting suggestion. We have now performed additional experiments of water supercooling where we use two binary alkane/alcohol mixtures, $C_5 + C_5OH$ and $C_6 + C_5OH$, with the concentration of alcohol (C_5OH) being 1% (v/v) to seal the water phase. The choice of C_5 and C_6 is due to the original high freezing frequencies observed when they are used as the sealing phase among all the alkanes. We tested the idea whether the small fraction of alcohols could generate an interfacial layer between the binary mixture and water, forming hydrogen bonds between alcohol and water molecules. This would, in theory as suggested, result in a low freezing frequency for the water, where results should reach that of the sealing by the pure C_5OH solution. Given that C_5OH is denser than both alkanes, one would expect it to be at the water interface forming a monolayer. However, we find that either such an interface is not stable or that it never fully forms and the interface consists likely as a mixture of alcohols

and alkanes. Neither of the alkane/alcohol binary mixtures are nearly as effective in suppressing freezing as the pure alcohols (Fig. S5(A)). In fact, the freezing frequency achieved by sealing via binary mixtures are not statistically different from the pure alkanes (Fig. S5(A)).

Surprisingly, in our experiments we observed that a significant amount of water will be sucked into the bulk of the binary alkane/alcohol mixture, forming emulsions (a milky/cloudy phase) on top of sealed water (Fig. S5(B)). An explanation or rather a starting point for explaining our observations in these tests might be found in the 2001 study of Stubbs et al [1]. In this study, via Monte Carlo simulations, they found that 1-hexanol segregates and forms clusters of 2-8 molecules in n-hexane rich binary mixtures. An experimental study by Gupta et al. [2] suggested that these clusters are polymer-like chains which form through cooperative hydrogen bonding. Overall these two studies suggest that alcohols in alkane rich binary mixtures form very heterogenous microdomains as opposed to a segregated monolayer. Given the similarity of our binary system to those studied in these two studies, we might start to understand why a pure alcohol-water interface is not likely in our experiments. The likely heterogeneity of the interface and further the heterogeneity of the bulk of the binary mixture itself might cause water molecules to be driven into this binary mixture. Nevertheless, further numerical and experimental investigations are certainly needed to investigate the exact mechanisms of this unexpected behavior and the water pumping into binary alkane/alcohol mixtures and the formation of the emulsions we observed. We have added Fig. S5 and its caption (Lines 198-204) in SI and Lines 168-172 in the main text.

We, once again, thank the reviewer for this very interesting suggestion and initiating experiments which brought about further questions that we hope to study in the near future.

3. Source of common oils. In order that others can repeat the experiment, it is suggested that the authors add, in the materials and methods section of SI, the sources of common oils used in this study, i.e., PO, MO, OO, and NO.

Response: We apologize for this omission. All of these chemicals were purchased from Sigma-Aldrich, USA. We have included this information in the section of “Materials and methods” in SI (Lines 8-11).

4. In captions for Figs. 3 and 4, note the volumes of water used for the results shown.

Response: We thank the reviewer for this suggestion. The water volume of each sample is 1 ml. We have included this information in the captions of Figs. 3 and 4 (Lines 513 - 514 and 522 in the main text).

Reviewer #2:

In this paper, Huang et. al. measured the cumulative freezing frequency of water (>1 ml) in presence and absence of different oils on water surface. The results show remarkable decrease of freezing frequency in presence of oils. Since the oils contain a mixture hydrocarbons and even

fatty acid, the authors have examined the change in freezing frequency in presence of pure oil-like compounds, such as linear alkanes and alcohols; and the results are qualitatively similar to that of the oils. Based on these results, the authors made an attempt to provide a molecular level picture of the hindered ice nucleation at the oil-water interface. The results are interesting, and may be publishable in Nat. Comm. after consideration of the following comments:

1. In the abstract it is mentioned that, “The relationship of this capacity with the chain length, however, shows opposite trends for alcohols and alkanes due to their drastically different interfacial structures with the water molecules.” Similar statements are also there in the main text. However, if we see Figure 3A which is the basis of the above statements, the opposite behavior is not so obvious. For example, in the case of alkane, on increasing the no of carbon from 5 to 8, ff decreases by ~ 50%. But in the case of alcohol, the ff increases only by ~ 5%. Now given the error bar of the results, it is reasonable to believe that in case of alcohols (except butanol which is much more soluble in water than the other alcohols examined) ff has already reached its optimum value even with c5-alcohol. Interestingly, the ff values with alcohols (c5 to c8) are close to the ff value of C10 or c11 alkanes. Thus, it seems to us that the alcohols are more efficient than alkanes in inhibiting ice nucleation at interface, at least in the regime of short chains (~ C<10).

Response: We thank you very much for this careful examination and insightful suggestion.

We agree absolutely that except butanol (C₄OH), C₅OH is the best sealing agent for deep supercooling among alcohols according to Fig. 3. We also agree with the reviewer that in the regime of short chain (carbon number m < 9) sealing agents, alcohols have higher freezing inhibition capacity than counterpart alkanes. When the carbon number m ≥ 9, the alcohols themselves can be frozen at -20 °C to form starkly different crystalline structure at the interface, triggering DSC water to freeze. This is also the reason why we have studied alcohols with 4 ≤ m ≤ 8, as illustrated in the caption of Fig. 3.

We agree that, compared to alkanes from C₅ to C₈, alcohols with the same carbon number and chain length (from C₅OH to C₈OH) have less variation in terms of absolute freezing frequency f_f (72.8% — 29.9% for C₅ — C₈ vs. 12.8% — 21.4% for C₅OH — C₈OH at Day-1). Nevertheless, the relative changes in average f_f for both alkanes and alcohols are significant (58.9% relative decrease of f_f from C₅ to C₈ and 67.2% relative increase of f_f from C₅OH to C₈OH). We, however, agree with the reviewer that if butanol is excluded from the statistical analysis, a trend for the alcohols do not exist, that is to say, the f_f of C₅OH—C₈OH are statistically similar. Accordingly, we have now removed the assertion (Lines 14 – 16, 203 – 207, and 305) that we observed opposite trends for alkanes and alcohols with respect to chain length from the manuscript. A revised and more detailed discussion has been presented in the main text (Lines 203 – 207 and 305) with the inclusion of C₄OH. We further detail our reasoning below for this inclusion.

Although C₄OH has relatively high solubility in water (section of “Freezing point depression due to oil-water mixing” in SI), its significantly lower f_f (4.4%) is more likely due to the ordering effect at water-oil interface than freezing point depression due to its solubility (which is predicted to contribute a maximum $\Delta T_F = 1.82$ °C, in contrast to $\Delta T = 20$ °C, please see section of “Freezing point depression due to oil-water mixing” in SI). As

a result, even though the absolute difference of f_f of DSC water sealed by alcohols C_5OH and C_8OH is not so big as that sealed by corresponding alkanes C_5 and C_8 , the overall different trends of f_f with regards to the chain length of alkanes and alcohols could be concluded. We have included these explanations in the revised manuscript (Lines 14 – 16, 203 – 207, and 305).

2. To explain the differences in f_f values between alkane and alcohol, the authors considered different structure and orientation of the alkyl chains as well as that of the water at the interface. The reasons of different behavior of alkane and alcohol presumably lie with the different interfacial structure. However, the molecular level understanding of the alkane-water and alcohol-water interfaces are not well established yet, experimentally. In fact recently SFG studies have suggested negative charging of the aqueous interface in presence of alcohol and alkane. As a result, the molecular level explanation of the alcohol and alkane induced freezing inhibition appears to be speculative. It needs further investigation to propose such a picture.

Response: We thank you for this insightful comment. We agree that neither a consensus nor a complete picture of molecular structures of alkane/water or alcohol/water interface have been well established through experimental measurements. Nevertheless, to hypothesize about a plausible mechanism for the freezing inhibition effects of surface sealing by alkanes and alcohols, we studied the existing experimental and numerical literature. This literature includes information on the interfacial microstructures, the change of microstructures about the chain length of alkanes and alcohols, and the relationship between ice nucleation and the microstructures.

For alkane/water interface, an electron deficit layer between alkane and water was discovered and confirmed by both X-ray reflectivity (XR) measurements [3-5] and atomistic molecular dynamics (MD) simulations [6, 7]. The preferentially parallel arrangement of alkane molecules along the interface with a large tilt angle β , and the increase of β with carbon number and chain length were also revealed by XR investigation and MD simulation [3, 7]. The mechanisms of ice nucleation on hydrophobic alkane interface via buckling and templating were illustrated by MD simulation [8]. Based on this knowledge, we hypothesized a plausible and likely mechanism for water freezing inhibition by alkane sealing as displayed in Fig. 3(B). Testing of this hypothesis requires further studies as suggested in our concluding paragraphs.

For alcohol/water interfaces, it is reasonable that no electron depletion layer or gap exists due to the hydrogen-bonding between hydroxyl group of alcohols and water molecules. According to the measurements by grazing incidence X-ray diffraction (GIXD) [9, 10] and MD simulation [11], the alcohol molecules would preferentially align themselves perpendicular to the interface with small tilt angle β due to the competition of the formation of hydrogen bonding. In addition, the longer the alcohol chains, the smaller the tilt angle β would be, inducing better lattice match between hexagonal ice and ordered alcohol layer at the interface, higher ice nucleation probability, and higher freezing temperature again based on consistent GIXD measurements and MD simulation [10, 11]. Furthermore, shorter alcohols would have stronger in- and out-of-plane fluctuations of their hydrogen bonds at the interface than longer alcohols, destabilizing templating for ice nucleation according to

MD simulation [11]. Taken together, we hypothesized Fig. 3(C) to explain the phenomena of freezing inhibition enabled by surface sealing with amphiphilic alcohols. In addition, these proposed microstructures of alkane/water and alcohol/water interfaces and their mechanisms on freezing inhibition are consistent with the physical interpretation from the perspective of thermodynamics (Lines 197-202 and 235-240). We revised the main text in Lines 193 - 194, 210, and 218 - 228 to clarify these proposed mechanisms.

Surface sealing by alkanes or alcohols might induce an electric field on the interfacial water. The electric field could also contribute to the inhibition or promotion of ice nucleation [12]. But the direction of electrical field (“positive” or “negative”) and induced orientation of water molecules (“hydrogen down” or “hydrogen up”) are not consistent and even contradictory according to sum frequency spectroscopy (SFG) measurements and MD simulations [13-15]. Therefore, to establish a concrete and clear picture for the effects of imposed electric field by alkane or alcohol sealing requires further investigations. We included a summary of this discussion of electric field on water freezing in Lines 311 - 316.

3. Line 167- 170: Extrapolation of the alkyl chain length dependence to different kinds of oils (PO vs MO) might lead to an over analysis of the result. This because the oils are mixture of many components including fatty acids which is expected to have stronger effect on interfacial water than alkane.

Response: We thank you for this comment. We agree that oils are mixtures of hydrocarbons beyond just the alkanes and these other hydrocarbon chains might also contribute to the differences in freezing frequencies we observed between PO and MO. To the best of our knowledge, the fatty acid concentration in PO and MO are negligibly low as opposed to olive oil. At room temperature (25 °C), MO’s dynamic viscosities is 23 cP [16], while heavy paraffin oil (PO) has a viscosity of 34 cP, which strongly indicates larger carbon number and longer chain length of alkanes in PO. Therefore, we utilized both dynamic viscosity and density to illustrate the composition difference between MO and PO in the revised manuscript (Lines 182 - 184). Although the variations of freezing inhibition capacity of MO and PO cannot be regarded as a direct consequences of different carbon number and chain length as there are surly some common alkanes in both oils, it might be regarded as a clue for the potential relationship between the capacity of freezing inhibition and the chain length of sealing agents.

We have revised the text (Lines 175-182) to reduce the emphasis in attributing this difference to the chain length of alkanes by itself and discussed the complex nature of the oils in this context. We noted that the different freezing inhibition capacities between PO and MO might only be partially explained via alkane chain length; and other hydrocarbon groups (which we did not study yet) might also contribute to the differences.

4. Line 82: the homogeneous ice nucleation has been explained by the energy terms of the triple interface as a form of Young equation (Line 82 in the text) and the corresponding model is shown in Figure S1A. Is the Figure consistent with the above equation? The author should show the derivation of the equation (in Supporting information) by the vector component analysis.

Response: We thank you for this suggestion. We have now revised the SI as suggested as follows:

To illustrate the forces at the triple interface (ice/water/air or ice/water/oil) during ice nucleation, the vectors of interfacial tensions were depicted as shown in the inset of Fig. S1(A). According to the force balance of vector components at the tangential ($x -$) direction, the Young's equation can be obtained as demonstrated in the newly added section of "Derivation of Young's equation at triple interface" in SI. Therefore, Fig. S1 and the vector components of interfacial tensions are consistent with the Young's equation in the main text. We also revised the caption of Figure S1 (Lines 183-185) in SI.

Reviewer #3:

This work reported long-term super-cooling of a few millimeter of pure water at temperatures as low as -20 C. This was achieved by sealing water with immiscible oil or alcohol. The authors also tried to explain the observed phenomena based on theoretical analyses of interfacial tension. Although the observed phenomena is somewhat interesting, it is unclear if the phenomena could still be observed if pure water is replaced with more biologically relevant samples including saline, cell culture medium, and suspensions of cells, tissues, and organs. Therefore, the manuscript as it stands currently is more appropriate for a physics journal than Nature Communications. My major and minor concerns on this study are given below:

Major

1. The authors mentioned in multiple places in the manuscript that the reported super-cooling phenomenon might be applied for the preservation of cells, tissues, and organs. However, no data is provided to support this statement at all. This is a major concern, because the equilibrium in the super-cooled water may not be maintained long if there are impurities in the water according to the authors (lines 122-124). For preservation of cells/tissues/organs, the sample is not pure water but water with impurities (i.e., salts, proteins, cells, tissues, or organs). Therefore, without convincing data on long-term preservation of organs, tissues, or at least cells, this work is more appropriate for publication in a physics journal.

Response: We thank you very much for this suggestion. Since our lab is heavily involved in biopreservation, the translation of this work into that realm is critical to our work.

We agree that the impurities, such as salts, proteins, and cells, might serve as catalysts for ice nucleation during supercooling of aqueous solutions or biospecimens, especially during long-term biopreservation. Therefore, we tested the efficacy of surface sealing by heavy paraffin oil (PO) to achieve DSC of suspensions of human red blood cells (hRBCs) in two solutions, anticoagulant citrate phosphate double dextrose supplemented with additive solution 3 (CP2D + AS-3) and University of Wisconsin solution supplemented with trehalose (UW+Tre). Successful DSC was achieved for these two cell suspensions at -7 °C, -13 °C down to -16 °C for as long as 100 days (Fig. 5 and S6). We did not observe significant ice formation until the lowest -16 °C (f_i is roughly 50% at -16 °C and smaller than 5% at -13 °C or above, Fig. S6). The f_i is almost the same on 42nd and 100th days, which is consistent with previous observations that no significant freezing occurs after certain period (usually 3 days, Figs. 3, S2, and S3). Compared to the f_i of about 0-10% for pure water for a period of 7 days in Fig. S2, the results at -16 °C indicate an increase in freezing

frequency. Nevertheless, at $-13\text{ }^{\circ}\text{C}$ or higher temperatures, there is no increase in f_f for these cell suspensions compared to pure water. Therefore, we can infer that salts, proteins, and cells in aqueous suspensions do not necessarily catalyze ice nucleation to the same extent at all temperatures. In our experimental observation window of -16 through $-7\text{ }^{\circ}\text{C}$, their addition only results in an increased freezing frequency at $-16\text{ }^{\circ}\text{C}$. Thus, surface sealing approach allows for the DSC at radically low supercooled temperatures to be achieved for cellular preservation, for up to 100 days.

In this preliminary translational experiment, we demonstrated improved biopreservation for hRBCs in UW+Tre at DSC temperatures with a drastically longer storage time, higher hemoglobin recovery rate, and less impaired morphology than conventional cold storage at $4\text{ }^{\circ}\text{C}$ in CP2D + AS-3 (Fig. 5). We added Figure 5, Figure S6-S7, and their captions into the manuscript, and accordingly, revised the title, abstract (Lines 19 - 22), main text (Lines 271 - 296), discussion (Lines 334 - 336), and the section of “Materials and methods” (Lines 52 - 75) in SI.

2. Lines 120-121, what are the unique and slightly different microstructures on the interface? Experimental data are needed to show the microstructures.

Response: We apologize for this misleading statement. The fact that some of the sealed water samples are more susceptible to freezing than others (“case-specific”) is probably because they have more impurities that catalyze ice nucleation. Therefore, we deleted these misleading and ambiguous statements in the revised manuscript.

3. Lines 191-192, experimental data are needed to support this statement.

Response: When water interfaces with amphiphilic alcohols, hydrogen bonds will form between water molecules and hydroxyl groups of alcohol molecules. In our study, primary linear alcohols are used to seal water surface, generating hydrogen bonds at one end of contacting alcohol molecules. The competition for the formation of hydrogen bonds at the alcohols/water interface to minimize the interfacial energy will cause perpendicular orientation of alcohol molecules as shown in Fig. 3(C). Both numerical simulation via molecular dynamics (MD) [17], and experimental study via grazing incidence X-ray diffraction (GIXD) [10], revealed small tilt angle ($\beta < 30^{\circ}$). Therefore, we revised our text by adding these references to support this statement (Line 210).

4. Line 208, extrapolating is not a reliable approach and experimental data are needed to support the extrapolation.

Response: We apologize about this ambiguous and somewhat misleading statement about extrapolation and references we omitted. Although molecular structures of alcohol/water interface are dominantly investigated for long alcohol chains (usually $m \geq 16$), experimental measurements via GIXD had reported tilt angles of shorter alcohols, $\beta = 28^{\circ}$ for $m = 6$ and $\beta = 30^{\circ}$ for $m = 5$ [10]. These tilt angles for the short chains are much bigger than those of long alcohols ($\beta < 20^{\circ}$ for $m \geq 16$), and thus demonstrate the general trend of tilt angle with respect to chain length when combined with the long-chain data, i.e., shorter alcohols have larger tilt angle β and bigger crystalline mismatches between ice nucleus and ordered alcohols. Accordingly, we are not extrapolating the trend into shorter alcohols

with $4 \leq m \leq 8$. Therefore, we believe that this trend is applicable to these short alcohols of similar structures in our study. We revised our statement in Lines 218 – 228 in the main text to clarify this confusion.

Minor

5. Lines 26-28, this is an exaggerated statement. I strongly disagree that the freezing and super-cooling of water are the least understood phenomena in our daily lives and scientific research. There is good amount of knowledge on this topic compared to many other topics including cell biology.

Response: We thank you for this suggestion. We have removed this statement in the revised manuscript.

6. Line 49, this statement is incorrect. Only pure water freezes at 0 °C under 1 atm, but it takes a long time.

Response: We apologize for the misunderstanding and the ambiguity in this statement. We agree that only pure water could freeze at 0 °C under 1 atm over a long time. Ice nucleation, a random process, can take a long time at (0 °C under 1 atm) which can be beyond the observation time in many experiments. Accordingly, the observed water freezing temperature can vary significantly in different experimental observations. However, the reverse process, ice melting, can stably occur at 0 °C under 1 atm. As a result, we utilized the stable melting temperature of ice as ice-water equilibrium temperature (T_e). We revised our statement to utilize this terminology to avoid this misunderstanding and removed the ambiguity in the main text (Lines 51 - 55).

7. Line 78, I do not see the symbol ϕ in the mathematical equation.

Response: We apologize for this misplacement. We deleted the definition of symbol ϕ here and added it later in Lines 91-92 where it first appears in the equation.

8. Lines 83-90, it is unclear why the contact angle for ice-water-oil should be nearly 0, since they are immiscible. In contrast, the contact angle for ice-water-air should be close to 0 because they are more miscible.

Response: We thank you for this question so that we can clarify our manuscript. The water contact angle at the ice/water/air or ice/water/oil triple interface is determined by the requirement of minimum free energy at interfaces [18], and can be obtained by the vector component analysis of interfacial tensions as shown in Fig. S1(A). Since oils are usually very immiscible or repellent to ice, water tends to completely wet ice nucleus, causing water contact angle on ice towards 0° [19, 20]. On the other hand, complete wetting of ice nucleus by water at ice/water/air interface does not generally occur (contact angle ~ 12°) [21]. Therefore, the water contact angle on ice embryo at ice/water/air interface is bigger than that at ice/water/oil interface. We revised Line 94 in the main text and added the section, “Derivation of Young’s equation at triple interface” (Lines 83-93) in SI to further clarify.

9. Lines 271-272, this statement is questionable. For example, PVA is a long chain alcohol and has been used as the ice nucleation inhibitor instead of initiating ice nucleation.

Response: We thank you for this comment so that we can clarify our manuscript and revise this incomplete statement. In this manuscript we only dealt with linear alkanes and linear primary alcohol chains which are immiscible with water such that they seal the water surface but don't mix with it. As such, our statement was meant to be limited to only linear primary alcohol chains which are immiscible with water; this implicit and incomplete statement was misleading as you have pointed out. Traditionally surface monolayers of long primary linear alcohols(C₂₈₋₃₁OH) have been used to initiate nucleation [9] and this was the contrast we wanted to point out. Unlike the primary linear alcohols, PVA can have many OH groups and is miscible with water and is thus excluded.

We have now revised the manuscript (Line 321) so that our implicit point is made explicit in the text to avoid any confusion.

1. Stubbs, J.M., et al., *Monte Carlo calculations for the phase equilibria of alkanes, alcohols, water, and their mixtures*. Fluid phase equilibria, 2001. **183**: p. 301-309.
2. Gupta, R.B. and R.L. Brinkley, *Hydrogen-bond cooperativity in 1-alkanol+ n-alkane binary mixtures*. AIChE journal, 1998. **44**(1): p. 207-213.
3. Fukuto, M., et al., *Nanoscale Structure of the Oil-Water Interface*. Phys Rev Lett, 2016. **117**(25): p. 256102.
4. Jensen, T.R., et al., *Water in contact with extended hydrophobic surfaces: direct evidence of weak dewetting*. Phys Rev Lett, 2003. **90**(8): p. 086101.
5. Mitrinović, D.M., et al., *Noncapillary-wave structure at the water-alkane interface*. Physical review letters, 2000. **85**(3): p. 582.
6. Wick, C.D., et al., *Computational investigation of the n-alkane/water interface with many-body potentials: the effect of chain length and ion distributions*. The Journal of Physical Chemistry C, 2011. **116**(1): p. 783-790.
7. Qiu, Y. and V. Molinero, *Strength of Alkane-Fluid Attraction Determines the Interfacial Orientation of Liquid Alkanes and Their Crystallization through Heterogeneous or Homogeneous Mechanisms*. Crystals, 2017. **7**(3): p. 86.
8. Fitzner, M., et al., *The Many Faces of Heterogeneous Ice Nucleation: Interplay Between Surface Morphology and Hydrophobicity*. J Am Chem Soc, 2015. **137**(42): p. 13658-69.
9. Gavish, M., et al., *Ice nucleation by alcohols arranged in monolayers at the surface of water drops*. Science, 1990. **250**(4983): p. 973-5.
10. Popovitz-Biro, R., et al., *Induced freezing of supercooled water into ice by self-assembled crystalline monolayers of amphiphilic alcohols at the air-water interface*. Journal of the American Chemical Society, 1994. **116**(4): p. 1179-1191.
11. Qiu, Y., et al., *Ice Nucleation Efficiency of Hydroxylated Organic Surfaces Is Controlled by Their Structural Fluctuations and Mismatch to Ice*. J Am Chem Soc, 2017. **139**(8): p. 3052-3064.
12. Ehre, D., et al., *Water freezes differently on positively and negatively charged surfaces of pyroelectric materials*. Science, 2010. **327**(5966): p. 672-5.
13. Mondal, J.A., et al., *Alkyl Chain Length Dependent Structural and Orientational Transformations of Water at Alcohol-Water Interfaces and Its Relevance to Atmospheric Aerosols*. J Phys Chem Lett, 2017. **8**(7): p. 1637-1644.

14. Jensen, T.R., et al., *Water in contact with extended hydrophobic surfaces: Direct evidence of weak dewetting*. Physical review letters, 2003. **90**(8): p. 086101.
15. Brown, M.G., et al., *Vibrational sum-frequency spectroscopy of alkane/water interfaces: Experiment and theoretical simulation*. The Journal of Physical Chemistry B, 2003. **107**(1): p. 237-244.
16. Stan, C.A., S.K. Tang, and G.M. Whitesides, *Independent control of drop size and velocity in microfluidic flow-focusing generators using variable temperature and flow rate*. Analytical chemistry, 2009. **81**(6): p. 2399-2402.
17. Dai, Y. and J.S. Evans, *Molecular Dynamics Simulations of Template-Assisted Nucleation: Alcohol Monolayers at the Air– Water Interface and Ice Formation*. The Journal of Physical Chemistry B, 2001. **105**(44): p. 10831-10837.
18. Makkonen, L., *Young's equation revisited*. J Phys Condens Matter, 2016. **28**(13): p. 135001.
19. Tabazadeh, A., Y.S. Djikaev, and H. Reiss, *Surface crystallization of supercooled water in clouds*. Proc Natl Acad Sci U S A, 2002. **99**(25): p. 15873-8.
20. Konno, A. and K. Izumiyama. *On the relationship of the oil/water interfacial tension and the spread of oil slick under ice cover*. in *Proceedings of the 17th International Symposium on Okhotsk Sea & Sea Ice*. 2002.
21. Knight, C.A., *The contact angle of water on ice*. Journal of Colloid and Interface Science, 1967. **25**(2): p. 280-284.

REVIEWERS' COMMENTS:

Reviewer #1 (Remarks to the Author):

In the revised manuscript, the authors adequately addressed the concerns raised in my original review report, as well as most of the comments from the other referees. Moreover, their newly added results support the utility of their surface-sealing method in improving the preservation of human red blood cells over the existing method used in current practice. This raises the significance of the authors' work further. Considering all these factors, I concur with the acceptance of this work for publication in Nature Communications.

Just one minor suggestion—in lines 170-172, the authors have added the statement:

"..., pumping water into the mixture to produce milky emulsions on top of water that significantly disrupts ice nucleation and water freezing (Fig. S5)."

I suggest that the authors remove the part "that significantly disrupts ice nucleation and water freezing". This part may confuse the readers since it could be taken as something that could help inhibition of ice formation.

Reviewer #3 (Remarks to the Author):

This revised manuscript is significantly improved in terms of both quality and clarity. I have two more comments regarding the new data presented in this revised manuscript:

1. In response to my previous comments, the author added data on red cell preservation. However, red cells are much simple system compared to eukaryotic cells. The fact that the method works for red cells does not mean it may work for eukaryotic cells. Although I do recognize the capability of preserving red cells is significant, the author should make it clear in the title of the manuscript by changing "cell suspension" to "red cell suspension".
2. The volume of the red cell suspension for the preservation studies is 1 ml. However, the authors claim that the method works for large volume of up to 100 ml. Would you expect to have the same result for red cell preservation if the sample volume is 10 or 100 ml?

Response Letter to Reviewers (NCOMMS-18-06979A)

We thank you and all the reviewers for their final insightful comments, and very helpful suggestions. We believe that we have now tended to all these final outstanding issues below and in the main text. The main text has also been revised to adhere to the Nature Communications formatting and section guidelines as requested.

Best Regards,

Dr. Berk Usta

Reviewer #1 (Remarks to the Author):

In the revised manuscript, the authors adequately addressed the concerns raised in my original review report, as well as most of the comments from the other referees. Moreover, their newly added results support the utility of their surface-sealing method in improving the preservation of human red blood cells over the existing method used in current practice. This raises the significance of the authors' work further. Considering all these factors, I concur with the acceptance of this work for publication in Nature Communications.

Just one minor suggestion—in lines 170-172, the authors have added the statement: "..., pumping water into the mixture to produce milky emulsions on top of water that significantly disrupts ice nucleation and water freezing (Fig. S5)." I suggest that the authors remove the part "that significantly disrupts ice nucleation and water freezing". This part may confuse the readers since it could be taken as something that could help inhibition of ice formation.

Response: We thank you for this suggestion. We removed this part of the text in the revised manuscript to avoid potential confusions to readers.

Reviewer #3 (Remarks to the Author):

This revised manuscript is significantly improved in terms of both quality and clarity. I have two more comments regarding the new data presented in this revised manuscript:

1. In response to my previous comments, the author added data on red cell preservation. However, red cells are much simple system compared to eukaryotic cells. The fact that the method works for red cells does not mean it may work for eukaryotic cells. Although I do recognize the capability of preserving red cells is significant, the author should make it clear in the title of the manuscript by changing "cell suspension" to "red cell suspension".

Response: We thank you for this suggestion to clarify our work better in the title. We modified the relevant part of the title to "red cell suspension" to specify the cell type preserved in our work.

2. The volume of the red cell suspension for the preservation studies is 1 ml. However, the authors claim that the method works for large volume of up to 100 ml. Would you expect to have the same result for red cell preservation if the sample volume is 10 or 100 ml?

Response: We thank you for this comment. We would expect similar results for human red blood cell preservation when the suspension volume is increased to the magnitude of 10^1 even 10^2 ml. Actually, in our pilot experiments, 5 out of 6 vials of 30 ml red cell suspension (containing 500 million cells) kept unfrozen after 365-day storage at $-13\text{ }^{\circ}\text{C}$. The preservation quality of human red blood cells has not demonstrated any conspicuous correlation with suspension volume in this range as long as there is no successful ice nucleation or sample freezing. Therefore, we added this information in the revised manuscript in Lines 290-292.